# Diabetic Foot Osteomyelitis Caused by Co-Infection with Methicillin-Resistant *Staphylococcus aureus* and Multidrug-Resistant Extended-Spectrum ß-Lactamase-Producing *Escherichia coli*: A Case Report

Shiori Kitaya [1,2,*,†], Chieko Miura [3], Ayano Suzuki [3], Yoshimichi Imai [3], Koichi Tokuda [1] and Hajime Kanamori [1,*,†]

1 Department of Infectious Diseases, Internal Medicine, Tohoku University Graduate School of Medicine, Sendai 980-8575, Japan; tokuda@med.tohoku.ac.jp

2 Department of Otolaryngology, Head and Neck Surgery, Tohoku University Graduate School of Medicine, Sendai 980-8574, Japan

3 Department of Plastic and Reconstructive Surgery, Tohoku University Graduate School of Medicine, Sendai 980-8575, Japan; m-chieko@med.tohoku.ac.jp (C.M.); ayanojai@gmail.com (A.S.); yo-imai@med.tohoku.ac.jp (Y.I.)

* Correspondence: shiori.kitaya.b7@tohoku.ac.jp (S.K.); kanamori@med.tohoku.ac.jp (H.K.); Tel.: +81-22-717-7304 (S.K.); +81-22-717-7373 (H.K.)

† These authors contributed equally to this work.

**Abstract:** This case report describes a 47-year-old man with type 2 diabetes and its associated complications. The patient developed co-infection with methicillin-resistant *Staphylococcus aureus* (MRSA) and multidrug-resistant (MDR) extended-spectrum ß-lactamase (ESBL)-producing *Escherichia coli* following surgical amputation for osteomyelitis caused by diabetic foot infection (DFI). The patient had a history of recurrent hospitalization due to DFI and had received multiple antimicrobials. Intraoperative wound cultures identified MRSA and MDR ESBL-producing *E. coli* as the causative agents of the co-infection. Intravenous vancomycin and meropenem were administered. After surgery, daily debridement and hyperbaric oxygen therapy were performed. The patient underwent surgical wound closure and was discharged on day 86. Polymicrobial infections in DFIs worsen antimicrobial resistance, impede wound healing, and increase the risk of osteomyelitis and amputation. Furthermore, infections caused by MDR bacteria exacerbate challenges in infection control, clinical treatment, and patient outcomes. In DFI cases caused by co-infection with MDR bacteria, prompt and appropriate antimicrobial therapy, debridement, and regular wound care while considering transmission are essential.

**Keywords:** methicillin-resistant *Staphylococcus aureus*; multidrug-resistant extended-spectrum ß-lactamase-producing *Escherichia coli*; diabetic foot osteomyelitis; diabetic foot infections

## 1. Introduction

Osteomyelitis in patients with diabetic foot implies prolonged therapy, an increased need for surgery, a high recurrence rate, greater amputation risk, and lower treatment success [1]. The 5-year mortality rate of diabetic foot osteomyelitis (DFO) is approximately 50%, surpassing that of many cancers [2]. An increase in co-infections with antimicrobial-resistant (AMR) organisms has recently been reported in diabetic foot infections (DFIs) [3]. Polymicrobial infections exacerbate antimicrobial resistance, hinder wound healing, and increase the risk of amputation [4]. Moreover, multidrug-resistant (MDR) bacterial co-infections complicate infection control, clinical management, and patient outcomes [4]. We report a case of co-infection with methicillin-resistant *Staphylococcus aureus* (MRSA) and MDR extended-spectrum ß-lactamase (ESBL)-producing *Escherichia coli* following surgical amputation for osteomyelitis caused by DFI.

## 2. Detailed Case Description

A 47-year-old man visited the outpatient department of Tohoku University Hospital, Sendai, Miyagi, Japan, with a complaint of fever after prolonged walking. He had a history of frequent hospitalizations for DFI and had received various antimicrobials, including tazobactam/piperacillin, levofloxacin (LVFX), meropenem (MEPM), and minocycline. The patient had mild developmental disabilities, making it challenging for him to provide adequate foot care and control his diabetes (glycated hemoglobin [HbA1c] 9.8%). The DFI had progressed to osteomyelitis, requiring surgical amputation of the fourth and fifth toes from the Lisfranc joint. The patient experienced DFI recurrence after initial recovery and was diagnosed with recurrent osteomyelitis based on imaging (Figure 1a). Additional debridement was performed 26 days after admission. During debridement, wound cultures collected from the deep lesions revealed MRSA and ESBL-producing *E. coli* as the causative agents of the co-infection (Figure 1b). ESBL-producing *E. coli* was resistant to at least three antimicrobial categories, classifying it as MDR. DNA extraction was performed using a QIAamp DNA Mini Kit (Qiagen, Hilden, Germany). Library preparation and DNA fragmentation were performed on genomic DNA extracted from the sample using a Nextera DNA Flex Library Prep Kit (Illumina, San Diego, CA, USA). Nextera DNA CD Index was used as the index adapter, according to the manufacturer's instructions. Whole-genome sequencing (WGS) was performed on an Illumina iSeq 100 (Illumina) with paired-end 150 bp reads. WGS of MRSA and ESBL-producing *E. coli* isolated from deep wound specimen samples obtained during surgery revealed a novel sequence type (ST), 8494, for MRSA and confirmed ST1193 for *E. coli*, with putative virulence genes (MRSA: *lukS-PV*, *lukF-PV* [NCBI BioProject PRJNA015043], *E. coli*: *hlyA*, *iutA*, *fyuA*, *iroN*, *fim*, *pap*, *sfa*, *foc* [NCBI BioProject PRJNA015045]), and antimicrobial resistance genes (MRSA: *mecA*, *mecR*$_1$, *mecI* [NCBI BioProject PRJNA015043], *E. coli*: *bla*$_{\text{CTX-M-55}}$ [NCBI BioProject PRJNA015045]).

(a)                                   (b)

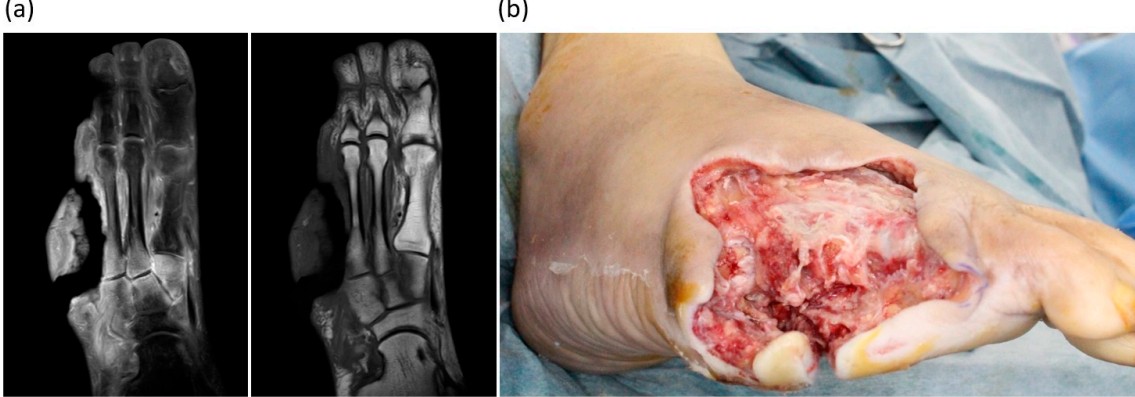

**Figure 1.** Magnetic resonance imaging and surgical findings in a patient with osteomyelitis caused by the exacerbation of a diabetic foot infection lesion due to co-infection with methicillin-resistant *Staphylococcus aureus* and multidrug-resistant extended-spectrum ß-lactamase-producing *Escherichia coli*. (**a**) The fourth and fifth toes of the right foot were amputated at the Lisfranc joint due to previous osteomyelitis. The base of the metatarsal bones of the third toe on the right foot showed high signal intensity on T2-weighted images and low signal intensity on T1-weighted images. This indicates the possibility of a new case of osteomyelitis. (**b**) The distal portion of the third toe on the right foot was additionally amputated due to osteomyelitis from the basal phalanx. Clearly necrotic tissue was completely removed, while ambiguous tissue was preserved. Methicillin-resistant *S. aureus* sequence type (ST) 8494 and multidrug-resistant extended-spectrum ß-lactamase-producing *E. coli* ST1193 were detected from cultures of deep wound specimens obtained during the surgery.

The patient was initially administered intravenous LVFX for 1 week, followed by oral administration for the next 3 weeks; the treatment was then changed to intravenous vancomycin and MEPM (1 g every 8 h for 6 weeks) based on the results of the wound culture

test (Figure 2). After surgery, the patient underwent daily debridement and received hyperbaric oxygen therapy (a total of 16 sessions, 60 min per session, at 2.5 atmosphere absolute). Surgical wound closure was performed on day 59, followed by negative pressure wound therapy for wound management (total of 16 days, suction pressure of −125 mmHg). The patient's treatment duration met the recommended treatment duration for osteomyelitis, and the patient was discharged on day 86 after satisfactory wound healing was confirmed by a reconstructive surgeon.

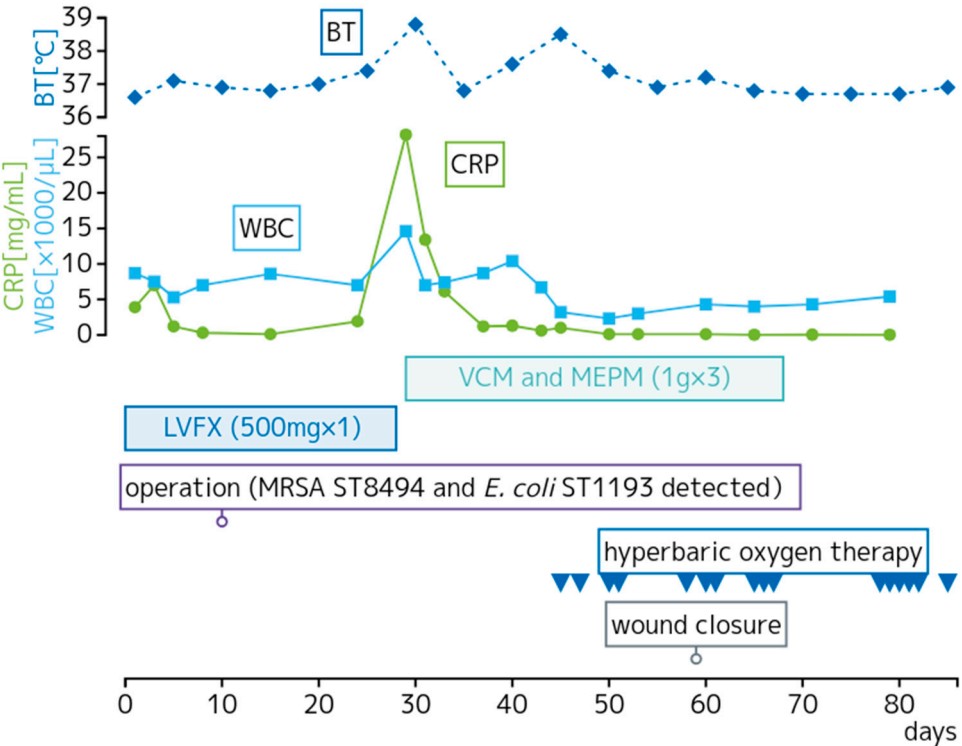

**Figure 2.** Vital signs and treatment progression during hospitalization. BT, body temperature; CRP, C-reactive protein; MEPM, meropenem; MRSA, methicillin-resistant *S. aureus*; LVFX, levofloxacin; ST, sequence type; VCM, vancomycin; WBC, white blood cell.

## 3. Discussion

### 3.1. Microbiological Characteristics of MRSA ST764 and E. coli ST1193

The ST764 strain is a hybrid variant of the ST5 lineage and is currently the most common MRSA genotype in Japan. It has been identified in many environments throughout Japan and is spreading worldwide [5,6]. Moreover, the ST764 strain is increasing in other Asian regions, such as China and Thailand [7,8]. This proliferation may be attributed to the presence of arginine catabolic mobile element type II, which could improve the bacteria's ability to colonize the skin and mucous membrane, thereby leading to more effective transmission [8]. The ST764 strain is frequently detected not only in the community but also in hospital environments, including outpatient departments and among healthcare professionals, and it has been reported as a possible cause of nosocomial MRSA infections [9]. The MRSA in this case was designated as novel ST8494, but the genetic difference between the ST764 strain and this new strain is limited to a single-point mutation in the *tpi* gene. Therefore, it is considered that the MRSA strain in this case has microbiological characteristics similar to those of the ST764 strain.

*E. coli* ST1193 is an emerging MDR high-risk clone that belongs to the highly pathogenic B2 lineage and is resistant to fluoroquinolones while producing ESBL [10]. ST1193 is the second most commonly isolated AMR *E. coli* clone after ST131 [10] and has been identified in many regions worldwide, including Japan [11–13]. Since 2012, its global prevalence has been on the rise, and in certain regions, it has replaced ST131 [10]. *E. coli* ST1193 and ST131

have prolonged gut persistence (over 6 months) and high rates of bacteriuria compared with those of other *E. coli* clones [14]. Furthermore, *E. coli* ST1193 is involved in both community-associated (CA) and hospital-associated (HA) urinary tract infections (UTIs) and bloodstream infections [11,12,15–17]. In addition, *E. coli* ST1193 has been reported to be responsible for HA-UTIs in long-term care facilities [11,12], as well as being implicated in sepsis and UTIs in children and neonates [18,19].

### 3.2. Virulence Genes of MRSA and MDR ESBL-Producing E. coli

The virulence of *S. aureus* is attributed to various surface components such as capsule polysaccharides, protein A, clumping factor, and fibronectin-binding protein, as well as extracellular proteins, including coagulase, hemolysins, enterotoxins, toxic shock syndrome toxin, exfoliatins, and Panton–Valentine leukocidin (PVL) [20,21]. Regarding *S. aureus* pathogenicity, there are three common virulence factors: toxic shock toxin-1 encoded by *tst*, PVL encoded by *pvl*, and a surface-targeting protein encoded by *sasX* [22]. In the present case, the presence of *pvl* was detected. *pvl* refers to a gene encoding a toxin responsible for tissue damage and immune system evasion and is produced by less than 5% of *S. aureus* strains [23]. Moreover, PVL creates pores in the mitochondrial membrane, destroying white blood cells, altering the immune system, and ultimately resulting in the dissolution and death of mitochondrial cell membranes [24,25]. The production of PVL is associated with necrotic lesions such as abscesses, subcutaneous tissues, severe CA-necrotizing pneumonia, disseminated infections, staphylococcal toxic shock syndrome, and osteomyelitis [26–29]. The *pvl* gene is detected in 93% of strains associated with pustular skin infections and in 85% of strains associated with severe necrotizing hemorrhagic pneumonia [30]. Patients infected with MRSA who test positive for PVL have a higher mortality rate [31]. Furthermore, previous studies have indicated a significantly higher prevalence of PVL in *S. aureus* strains associated with osteomyelitis compared to soft tissue infections [32]. Therefore, the presence of PVL could be considered one potential factor contributing to the progression of the severe clinical course from DFI to osteomyelitis in this case.

Uropathogenic *E. coli* (UPEC) is the most common extraintestinal pathogenic *E. coli* pathotype and is considered the leading cause of CA-UTIs and many HA-UTIs [33,34]. UPEC strains produce virulence factors encoded by pathogenicity islands, plasmids, and transposons. These factors can be classified into two categories: (1) secreted virulence factors (toxins and siderophore systems) and (2) cell surface-associated virulence factors (adhesins and invasins) [35]. The most crucial virulence factor secreted by UPEC is $\alpha$-hemolysin (HlyA), a pro-inflammatory toxin. HlyA is encoded by the *hlyA* gene in the pathogenicity island and enables bacteria to lyse erythrocytes, endothelial cells, and urinary tract cells, allowing the bacteria to capture iron and evade phagocytic cells [36,37]. This iron acquisition process is essential for UPEC's persistence, proliferation, and pathogenicity within the host [38]. HlyA also promotes the release of interleukin-6 (IL-6) and IL-8, contributing to infection severity [39]. Previous studies have suggested a potential association between HlyA production and severe infections such as sepsis and renal damage [40]. Furthermore, HlyA production is more frequent in UPEC strains causing pyelonephritis than in those causing cystitis, suggesting a link between HlyA and severe infections such as pyelonephritis [41]. The presence of *hlyA* gene showed a higher positivity rate in UPEC strains that were ESBL- and MDR-negative, contrasting with ESBL- and MDR-positive UPEC strains, indicating an association with low antimicrobial resistance in UPEC [42]. In this particular case, the detected *E. coli* was ESBL- and MDR-positive but possessed the *hlyA* gene.

The siderophore systems encoded by the *iutA*, *ireA*, *fyuA*, *iroN*, and *aer* genes enable *E. coli* to acquire iron from the host, facilitating its colonization and survival while protecting the bacteria from the toxic effects of this metal [43]. Siderophore systems have also been associated with the occurrence and severity of UTIs [44,45]. In this particular case, the strains possessed *iutA*, *fyuA*, and *iroN* genes. The *iutA* gene, carried on plasmids, is most frequently associated with strains exhibiting resistance to different antimicrobials, including

those containing antimicrobial resistance factors on the same plasmid [46]. The *iutA* gene is correlated with resistance to various antimicrobials, such as amoxicillin-clavulanic acid, ampicillin, cephalothin, cefotaxime (CTX), ceftazidime (CAZ), ciprofloxacin, gentamicin, tetracycline, and sulfamethoxazole/trimethoprim [46,47]. Positive associations between the presence of the *iutA* gene and multidrug resistance have been demonstrated [48]. Furthermore, other studies have reported that UPEC strains with the *iutA*, *fimH*, and *fyuA* genes show resistance to CTX and CAZ, suggesting a strong association between these three genes and resistance to third- and fourth-generation cephalosporins [49,50]. Reports indicate a correlation between antimicrobial resistance and decreased toxicity, with resistant strains showing a lower presence of toxicity genes [42]. However, in this particular case, although the strain was resistant, it possessed multiple toxicity genes, resulting in clinical progression to severe osteomyelitis.

The expression of surface adhesion factors enhances the virulence of pathogenic *E. coli* and initiates close contact between the bacteria and the host cell wall [51]. The most frequently detected adhesion factors include types 1, P, and S and F1C, encoded by the *fim*, *pap*, *sfa*, and *foc* operons, respectively [38,52]. In this case, all of these adhesion factors were detected. Types P, S, and 1 fimbriae are responsible for attachment to epithelial cells in the intestines, kidneys, or lower urinary tract, as well as stimulating cytokine production by T cells [53]. Additionally, they are crucial for colonization in extraintestinal infections [38]. The S-fimbriae adhesion factor was detected in this case, and it can bind to components of the extracellular matrix and sialoglycoproteins on brain capillary endothelial cells [53]. Therefore, adhesion factors are considered virulence factors present in strains that cause meningitis and sepsis and are believed to be associated with clinical severity [53]. Furthermore, a previous study reported that while the *fimH* gene is observed in osteomyelitis, it is not observed in infected skin and soft tissues, emphasizing the crucial role of this adhesive factor in facilitating the lesion's reach to the bone [54]. Considering the limited adhesive ability of *E. coli* to osteoblasts [55], the presence of the *fimH* gene in the *E. coli* strains in this specific case could have played a significant role in the progression from skin and soft tissue infection to severe osteomyelitis.

### 3.3. Antimicrobial Resistance Genes of MRSA and MDR ESBL-Producing E. coli

According to the Centers for Disease Control and Prevention's definition and antimicrobial susceptibility results, the MRSA strain in the present case was classified as HA-MRSA [56]. This classification is supported by the staphylococcal cassette chromosome mec type II in this strain, which encodes resistance to non-ß-lactam antimicrobials [57]. This HA-MRSA strain also carried *pvl* gene, typically associated with CA-MRSA [58,59]. Thus, this strain exhibited characteristics of both HA- and CA-MRSA. Frequent hospital admissions, medical procedures, and antimicrobial use may be associated with MRSA transmission between hospitals and the community [60]. Furthermore, hospital epidemiological surveillance has revealed suspected MRSA transmission among patients in the same room. In contrast, no evidence of nosocomial transmission of ESBL-producing *E. coli* existed.

$bla_{\text{CTX-M}}$ genes originate from the ß-lactamase genes of environmental bacteria, indicating a totally different origin from $bla_{\text{TEM}}$ and $bla_{\text{SHV}}$ genes, and could preferentially hydrolyze cefotaxime compared with TEM- and SHV-type enzymes [61]. $bla_{\text{CTX-M}}$ genes have been spreading as predominant ESBL types after the new millennium [62]. The situation has barely changed since 2010, and $bla_{\text{CTX-M-15}}$ genes dominate most regions worldwide [63]. Incidentally, the prevalence of $bla_{\text{CTX-M-55}}$ gene positivity is increasing in certain regions worldwide, especially in the South China region [64]. CTX-M-55 is a variant of CTX-M-15 with only one amino acid substitution (Ala-80-Val) [61]. Both CTX-M-15 and CTX-M-55 belong to the CTX-M-1 group, but CTX-M-55 exhibits high hydrolytic activity against ceftazidime [65]. A previous study revealed that among $bla_{\text{CTX-M-55}}$-positive *E. coli* isolates obtained from patients with UTIs, ST1193 (18%) was the most common ST, similar to the present case [19].

### 3.4. Clinical and Epidemiological Characteristics of DFI and DFO

Patients with diabetes are at risk of developing diabetic foot ulcers in approximately 19 to 34% of cases during their lifetime, and approximately 50% of these cases become infected [66,67]. Several studies have identified risk factors for the development of DFIs, including a mean duration of diabetic foot ulcers >30 days, trauma as the cause, wound extension to the bone, recurrent wounds, previous amputation surgery, peripheral arterial disease, loss of protective sensation, and renal failure [68–71]. In the current case, the patient exhibited all risk factors except a history of trauma. The pathogens involved in DFIs can vary depending on geographical region, socio-economic conditions, average ulcer duration, the depth of the ulcer, the presence of peripheral arterial disease, and other complicating factors [72–74]. *S. aureus*, *Streptococcus*, and *Enterococcus* are major Gram-positive pathogens, while Enterobacterales and *Pseudomonas aeruginosa* are well-known Gram-negative pathogens in DFIs [75]. In the present case, the causative agents were typical *S. aureus* and *E. coli*.

Chronic DFIs often lead to polymicrobial infections, particularly in patients who have received prior antimicrobial therapy [75]. Polymicrobial infections may facilitate the cross-transfer of resistant genes between different species, which could increase the risk of poor clinical outcomes in patients [4,76,77]. Polymicrobial infections and their synergistic interactions may lead to an increase in antimicrobial resistance, delayed wound healing, and an elevated risk of amputation in patients with diabetes [78,79]. The patient in the current case had poorly controlled diabetes as an underlying condition, and due to polymicrobial infection with AMR bacteria, DFI progressed to DFO, resulting in toe amputation.

As DFIs are typically chronic, prolonged and frequent use of antimicrobials is often required. Moreover, in systemic illness situations, initial broad-spectrum antimicrobials must be empirically administered before receiving results from microbial cultures, resulting in increased AMR bacteria incidence and antimicrobial resistance gene variations [80,81]. Studies have shown that over 70% of the causative bacteria in patients with DFIs are resistant to at least one antimicrobial agent, with more than half resistant to multiple agents [82] and 20% being MDR [73]. AMR bacteria such as MRSA and ESBL-producing bacteria are common causes of DFIs [73]. The main risk factors for multidrug resistance include previous antimicrobial therapy, prior amputation surgery, frequent hospitalizations, and chronic wound duration [83,84]. In the present case, the patient exhibited all of these risk factors. The incidence of co-infections caused by AMR bacteria is increasing, and one report suggests that 16.1% of patients carrying ESBL-producing Enterobacterales have coexistence or co-infection [3]. The diversity of co-infections caused by MDR bacteria further amplifies challenges in infection control, clinical treatment, and patient outcomes [7]. In the present case, the patient's history of frequent antimicrobial administration due to recurrent DFI contributed to polymicrobial infection with MRSA and MDR ESBL-producing *E. coli*, resulting in osteomyelitis.

DFO is the most common infection associated with DFI, occurring in approximately 20% of patients with mild infection and 50–60% of patients with severe infection [85–87]. The main mechanisms of DFO development involve either the hematogenous or contiguous spread of bacteria. On the one hand, in hematogenous osteomyelitis, bacteremia and the seeding of bones from distant sites are involved. On the other hand, contiguous spread involves direct inoculation into adjacent bone tissues, open fractures, penetrating injuries, or nosocomial contamination [88,89]. In the current case, it is believed that the primary mechanism of DFO development was contiguous spread through direct inoculation into adjacent bone tissues. DFO is usually related to advanced peripheral neuropathy, often accompanied by peripheral arterial disease, foot deformities, and suboptimal patient compliance with foot care recommendations [2]. In the current case, the factors leading to severe osteomyelitis and subsequent toe amputation included insufficient preventive foot care due to mild intellectual disability, inadequate diabetes management, and excessive pressure on the deformed toes due to high body weight. In patients with DFO, long-term antimicrobial therapy or amputation is often necessary to remove the infected bone [90]. Patients

with DFO have an approximately 20-fold amputation risk than those without DFO [91]. In the current case, treatment primarily involved approximately 6 weeks of appropriate intravenous antimicrobial administration and toe amputation. Regular debridement and wound care procedures are crucial to preventing such complications. Previous research has also highlighted cases in which wound procedures performed by healthcare providers inadvertently transmitted bacterial infections [92]. Therefore, caution should be exercised while managing wounds.

## 4. Conclusions

The patient in the present case developed DFO in DFI due to factors such as insufficient preventive foot care resulting from mild intellectual disability, inadequate diabetes management, and excessive pressure on the deformed toes due to high body weight. Treatment involved prolonged antimicrobial administration and toe amputation. Polymicrobial infections in DFI can increase the risk of adverse clinical outcomes, including delayed wound healing and elevated amputation risk. Moreover, the chronic nature of DFI often requires prolonged and frequent antimicrobial use, which may lead to variations in antimicrobial resistance genes and increased AMR incidence. Prompt and appropriate antimicrobial therapy and surgical interventions are essential for the management of DFI, especially in cases of co-infection with MDR bacteria, as the choice of treatment becomes crucial. In cases of DFI caused by co-infection with MDR bacteria, prompt and appropriate antimicrobial therapy, sufficient wound debridement, and regular wound care considering transmission are necessary.

**Author Contributions:** S.K.: conceptualization, methodology, data curation, data analysis, writing—original draft, writing—review and editing. C.M.: writing—review and editing. A.S.: writing—review and editing. Y.I.: writing—review and editing. K.T.: writing—review and editing. H.K.: conceptualization, methodology, investigation, writing—review, and editing. S.K. and H.K. contributed equally to this study. S.K. obtained written informed consent from the patient to publish the case report. All authors contributed to patient clinical management and reviewed the report. All authors have read and agreed to the published version of the manuscript.

**Funding:** This study was partially funded by joint research between Tohoku University and NBC Meshtec Inc.

**Institutional Review Board Statement:** The study was conducted in accordance with the Declaration of Helsinki, and approved by the Ethics Committee of Tohoku University Graduate School of Medicine (protocol code: 2019-1-270 and date of approval: 26 July 2019).

**Informed Consent Statement:** Informed consent was obtained from subject involved in the study.

**Data Availability Statement:** The data underlying the case report cannot be shared publicly to protect the privacy of the patient.

**Acknowledgments:** We would like to thank Yumiko Takei from the Department of Infectious Diseases, Internal Medicine, Tohoku University Graduate School of Medicine, Sendai, Japan, for her technical assistance with analyzing bacterial isolates.

**Conflicts of Interest:** The authors declare no conflict of interest.

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
