# Peer review of "Diabetic Foot Osteomyelitis Caused by Co-Infection with Methicillin-Resistant Staphylococcus aureus and Multidrug-Resistant Extended-Spectrum ß-Lactamase-Producing Escherichia coli: A Case Report"

_2673-8007, doi:10.3390/applmicrobiol3030072_

Round 1

Reviewer 1 Report

The article presents a simultaneous infection in a patient with diabetic foot. The multi-resistant bacteria is commonly found after prolonged/repeated hospital admissions, repeated antibiotics usage, especially in diabetes patients

The article could be rewritten as the outcome on certain antibiotics with focusing on antibiotics usage (Letter to editor)

Author Response

The article presents a simultaneous infection in a patient with diabetic foot. The multi-resistant bacteria is commonly found after prolonged/repeated hospital admissions, repeated antibiotics usage, especially in diabetes patients

The article could be rewritten as the outcome on certain antibiotics with focusing on antibiotics usage (Letter to editor)

Response: We thank the reviewer for this comment. In this case, treatment was administered as appropriate doses of vancomycin and meropenem based on the antimicrobial susceptibility results of MRSA and MDR ESBL-producing E. coli detected from the wound site. Furthermore, the treatment duration met the recommended period for osteomyelitis (6 weeks), and a thorough examination by a reconstructive surgeon confirmed satisfactory wound healing. We have documented the details of the antibiotics used in this case, including their content, dosage, and duration of administration, as well as meeting the treatment period requirements and confirming successful wound healing, in Lines 71–74 and Page 2, as well as Lines 78–80.

“The patient was initially administered intravenous levofloxacin for 1 week, followed by oral administration for the next 3 weeks; the treatment was then changed to intravenous vancomycin and meropenem (1 g every 8 h for 6 weeks) based on the results of the wound culture test (Figure 2).”

“The duration of treatment of the patient met the recommended treatment duration for osteomyelitis, and the patient was discharged on day 86 after confirming satisfactory wound healing by a reconstructive surgeon.”.

Reviewer 2 Report

Abstract: ok but lack much pieces of information. 

The whole genome sequencing of MRSA and ESBL-E. coli details must be attached as a supplementary file for easy understanding. 

The author reveals the novel sequence ST 8494 for MRSA and ST1193. It was also required to add figure show these novel sequences depicting putative virulence genes (MRSA: lukS-PV, lukF-PV [NCBI 66 BioProject PRJNA015043], E. coli: hlyA, iutA, fyuA, iroN, fim, pap, sfa, foc [NCBI BioProject 67 PRJNA015045]), and antimicrobial resistance genes (MRSA: mecA, mecR1, mecI [NCBI Bi- 68 oProject PRJNA015043], E. coli: blaCTX-M-55 [NCBI BioProject PRJNA015045]).

It is unclear whether the author detected these MRSA ST764 and E. coli ST1193 in their sample. if yes, then this must be depicted in a table or figure. 

How, in this particular 173 case, although the strain was resistant, it possessed multiple toxicity genes, resulting in 174 clinical progression to severe osteomyelitis: Authors are supposed to give details also. 

Authors are supposed to give details on how they have detected the antimicrobial resistance genes of MRSA and MDR ESBL-producing E. coli

No comments for English language. 

Author Response

The whole genome sequencing of MRSA and ESBL-E. coli details must be attached as a supplementary file for easy understanding.

Response: We thank the reviewer for this comment. Regarding the results of whole genome sequencing for MRSA and ESBL-E. coli, detailed information about antimicrobial resistance genes and virulence genes has been provided as supplementary files.

The author reveals the novel sequence ST 8494 for MRSA and ST1193. It was also required to add figure show these novel sequences depicting putative virulence genes (MRSA: lukS-PV, lukF-PV [NCBI 66 BioProject PRJNA015043], E. coli: hlyA, iutA, fyuA, iroN, fim, pap, sfa, foc [NCBI BioProject 67 PRJNA015045]), and antimicrobial resistance genes (MRSA: mecA, mecR1, mecI [NCBI Bi- 68 oProject PRJNA015043], E. coli: blaCTX-M-55 [NCBI BioProject PRJNA015045]).

?

Response: We thank the reviewer for this comment. We have provided detailed information regarding antimicrobial resistance genes and virulence genes for the detected MRSA and ESBL-E. coli in this case as supplementary files.

The S. aureus detected in this case was found to have only one differing base in the tpi gene, which is one of the housekeeping genes, from the existing ST764 strain. As a result, it was considered to be closely related to the ST764 strain. Based on the allele numbers of each housekeeping gene, we have attached a copy of the email containing the analysis results from the website where the actual analysis of the new ST was conducted.

It is unclear whether the author detected these MRSA ST764 and E. coli ST1193 in their sample. If yes, then this must be depicted in a table or figure.

Response: We thank the reviewer for this comment. To clearly illustrate the detection of MRSA ST764 and E. coli ST1193 from the actual surgical wound site, the legend of Figure 1 has been revised as follows:

“Methicillin-resistant S. aureus sequence type8494 and multidrug-resistant extended-spectrum ß-lactamase-producing E. coli ST1193 were detected from cultures of deep wound specimen obtained during the surgery.”

Furthermore, in Figure 2, we have added a note indicating the detection of MRSA ST764 and E. coli ST1193 from the actual surgical wound site during the surgery.

How, in this particular 173 case, although the strain was resistant, it possessed multiple toxicity genes, resulting in 174 clinical progression to severe osteomyelitis: Authors are supposed to give details also.

Response: Thank you for your feedback. As pointed out by the reviewer, both MRSA and MDR ESBL-producing E. coli in this case harbored multiple virulence factors. Particularly, the pvl gene in MRSA and the fimH gene in E. coli are considered to potentially play crucial roles in the progression to osteomyelitis. Therefore, to address this issue, we have added the following discussion on Page 4, Lines 140–143 and Page 5, Lines 192–198, respectively.

“Furthermore, previous studies have indicated a significantly higher prevalence of PVL in S. aureus strains associated with osteomyelitis compared to soft tissue infections [32]. Therefore, the presence of PVL could be considered as one potential factor contributing to the progression of severe clinical course from DFI to osteomyelitis in this case.”

“Furthermore, a previous study reported that while the fimH gene is observed in osteomyelitis, it is not observed in infected skin and soft tissues, emphasizing the crucial role of this adhesive factor in facilitating the lesion's reach to the bone [54]. Considering the limited adhesive ability of E. coli to osteoblasts [55], the presence of the fimH gene in the E. coli strains of this specific case could have played a significant role in the progression from skin and soft tissue infection to severe osteomyelitis.”

Authors are supposed to give details on how they have detected the antimicrobial resistance genes of MRSA and MDR ESBL-producing E. coli

Response: We thank the reviewer for this comment. We have added additional information regarding the detection methods for antimicrobial resistance and virulence genes in MRSA and MDR ESBL-producing E. coli as part of the Supplementary Methods, as follows:

“Antimicrobial resistance and virulence genes of methicillin-resistant Staphylococcus aureus and multidrug-resistant extended-spectrum ß-lactamase-producing Escherichia coli were examined using the online database EzBioCloud TruBac ID [3]. ”

Reviewer 3 Report

The authors present aa patients with DFI history who was hospitalized and diagnosed with MDR osteomyelitis that required partial foot amputation. This is a well-presented case with a nice discussion.

I hHave a few points for clarity:

1.      Table 1 perhaps needs to be called Figure 2?

2.      I would suggest to call it “surgical wound closure” to be unambiguous.

3.      How many sessions of HBOT? What was the final pressure and time at pressure?

4.      How long was NPWT applied? Suction pressure?

Author Response

The authors present aa patients with DFI history who was hospitalized and diagnosed with MDR osteomyelitis that required partial foot amputation. This is a well-presented case with a nice discussion.

I have a few points for clarity:

  1. Table 1 perhaps needs to be called Figure 2?

Response: Thank you for your feedback. We have corrected the title of Table 1 to Figure 2.

  1. I would suggest to call it “surgical wound closure” to be unambiguous.

Response: We thank the reviewer for this comment. Following the suggestions of Reviewer 2, we have made the following revisions to the abstract on Page 1, Lines 23–24, and the content in the main text on Page 2, Lines 76–78, respectively:

“The patient underwent surgical wound closure and was discharged on day 86.”

“Surgical wound closure was performed on day 59, followed by negative pressure wound therapy for wound management (total of 16 days, suction pressure of −125 mmHg).”

  1. How many sessions of HBOT? What was the final pressure and time at pressure?

Response: We thank the reviewer for this comment. We have added the following information regarding the number of attempts, duration, and pressure of HBOT in the main text on Page 2, Lines 74–76:

“After surgery, the patient underwent daily debridement and received hyperbaric oxygen therapy (a total of 16 sessions, 60 min per session, at 2.5 atmosphere absolute).”

  1. How long was NPWT applied? Suction pressure?

Response: We thank the reviewer for this comment. We have added the following information regarding the trial period and pressure of NPWT in the main text on Page 2, Lines 76–78:

“Surgical wound closure was performed on day 59, followed by negative pressure wound therapy for wound management (total of 16 days, suction pressure of −125 mmHg).”